# The Impact of Different Types of Rice and Cooking on Postprandial Glycemic Trends in Children with Type 1 Diabetes with or without Celiac Disease

**DOI:** 10.3390/nu15071654

**Published:** 2023-03-29

**Authors:** Antonio Colasanto, Silvia Savastio, Erica Pozzi, Carlotta Gorla, Jean Daniel Coïsson, Marco Arlorio, Ivana Rabbone

**Affiliations:** 1Department of Pharmaceutical Sciences, University of Piemonte Orientale, 28100 Novara, Italy; 2Division of Pediatrics, Department of Health Sciences, University of Piemonte Orientale, V. Solaroli 17, 28100 Novara, Italy

**Keywords:** rice, glycemic index, cooking, AHCL, children, type 1 diabetes

## Abstract

The aims of this study were to evaluate: (i) the chemical and nutritional composition of rice before and after cooking and (ii) postprandial glycemic impacts in children and adolescents with type 1 diabetes (T1D) after eating two different types of rice (“Gigante Vercelli” white rice and “Artemide” black rice) or white rice cooked “risotto” style or boiled using an advanced hybrid closed loop (AHCL) system (Tandem Control-IQ^TM^). General composition and spectrophotometric analyses of raw and cooked rice were performed. Eight T1D subjects (four males and four females, aged 11 ± 1.4 years), two with celiac disease (CD), using an AHCL system were enrolled. “Gigante Vercelli” white rice cooked as risotto or boiled and boiled “Artemide” rice were prepared by the same cook on two evenings. Continuous glucose monitoring metrics were evaluated for 12 h after meal consumption. Total dietary fiber was higher for both rice types after cooking compared with raw rice. Cooking as risotto increased polyphenols and antioxidants (*p* < 0.05) in both rice varieties, and total starch decreased after boiling (*p* < 0.05) in white rice. There was a significant peak in glycemia after consuming risotto and boiled white rice (*p* < 0.05), while the mean glycemic peak remained <180 mg/dL in individuals eating boiled Artemide rice. There were no significant differences in automatic basal or auto-bolus insulin deliveries by the AHCL according to different types of rice or cooking method. Our findings suggest that glycemic trends are impacted by the different chemical and nutritional profiles of rice but are nevertheless well controlled by AHCL systems.

## 1. Introduction

Rice (*Oryza sativa* L.) is not only a staple food in many diets worldwide but a popular gluten-free grain for people with celiac disease (CD). Rice can be bought as different varieties (white, brown, or pigmented) [1], colors (white, red, or black, depending on the pigment contents), caryopsis shape, and texture.

White rice varieties are currently the most cultivated and consumed, contrary to pigmented ones, whose cultivation is limited to restricted areas of the globe (including Italy, France and north Africa).

White rice (milled rice where the husk, bran, and germ have been removed) is primarily composed of starch, hypoallergenic proteins, and bioactive components (including fiber and tocopherols) and offers high levels of energy and glycemic index. However, unmilled rice (brown or pigmented) has a lower starch content and higher nutritional composition, in particular of proteins, fiber, vitamins, and minerals, so it is generally regarded as healthier and has a lower glycemic index. Furthermore, pigmented varieties such as Artemide black rice are generally richer in polyphenols, antioxidants, and anthocyanins than white varieties [2,3].

Italy (particularly the area between the Piedmont and Lombardia Regions, including the provinces of Novara, Vercelli and Pavia) is the principal rice producer in Europe, maintaining a large number of registered varieties characterized by different nutritional and technological features.

The Gigante Vercelli variety is an old Italian rice variety released in 1946 belonging to the japonica group that exhibits a considerable, wide, and durable resistance to many isolates of the blast fungus *Magnaporthe oryzae* that has lasted for more than 65 years [4]. This variety has not been cultivated for decades but was already selected as a donor of blast resistance in programs [5,6].

The rediscovery of Gigante Vercelli rice for the human diet started in 2017 with the work of a small number of rice farmers. The variety presents interesting nutritional properties mainly due to a significant content of resistant starch, important for digestion and intestinal health. Another important characteristic is the good gastronomic quality, showing a high resistance to cooking thanks to the high amylose content, especially when used for the preparation of risotto, a quality that makes it very similar to the Carnaroli rice from which it differs by requiring less cooking time [7].

The pigmented rice varieties (black, red and purple) may be pointed out as natural functional foods, considering their content of different phenolic classes and their significant antioxidant properties. The anthocyanin pigments of these rice varieties are mainly concentrated in the external layers of the bran, so their consumption is reserved for the unmilled form.

Black rice is mainly produced in Southeast Asia, but in recent years its cultivation in Italy has been gaining ground because of a greater culinary appreciation by the Italian population. The main varieties of black rice grown in Italy, and in particular in the Piedmont region, are Artemide, Venere and Nerone [8]. Artemide black rice is a pigmented variety obtained from the hybridization between Venere black rice and a white indica variety; it is particularly rich in healthy polyphenols, mainly anthocyanins and phenolic acids.

CD and type 1 diabetes (T1D) are both autoimmune diseases caused by immunological mechanisms resulting in villous and pancreatic beta cell damage respectively [9]. The prevalence of CD ranges from 1% to 16% in type 1 diabetes patients according to different studies and increases with a longer duration of diabetes [10]. Most cases of celiac disease are diagnosed within the first year of diabetes diagnosis, but the diagnosis can be made as late as adulthood and the association is common to both genders. Celiac disease is often asymptomatic and is not necessarily associated with poor growth, deterioration of glycemic control, or hypoglycemia. Screening for CD is recommended at diabetes onset and after 2 and 5 years, according to the International Society for Pediatric and Adolescent Diabetes Consensus Guidelines [10].

Diet and insulin therapy are critically important in the management of type 1 diabetes to improve glycemic control and reduce the risk of complications. However, when there is comorbid celiac disease, diabetes management can be complicated by the introduction of a gluten-free diet (GFD). The GFD normalizes the intestinal mucosa and often leads to the disappearance of antibodies but does not necessarily have an impact on glycemic control. Children and adolescents with T1D, with poor adherence to a gluten-free diet, may also have a reduced quality of life and poorer glycemic control [10,11,12]. Moreover, a GFD can be difficult to manage to maximize glycemic control with high glycemic index (GI) foods. While some studies have shown that a GFD can have a positive impact on children’s growth, data on the impact of a GFD on glycemic control, HbA1c, and hypoglycemic episodes in subjects with T1D and CD are conflicting. A low GI diet can certainly be useful for reducing glucose excursions, but most GFD foods have a high GI that can rapidly raise blood glucose levels and risk major glycemic variability [11].

Some food types, such as pizza and rice, are very difficult to manage in terms of glycemic response. Since rice has a high GI (58–93%), T1D children often struggle to control their postprandial glycemic levels after eating it. The GI is influenced by various intrinsic characteristics of the food such as starch content (amylose/amylopectin ratio), postharvest processing, and consumer processing (cooking, storage and heating) [13,14]. Continuous glucose monitoring (CGM) systems and advanced hybrid closed-loop (AHCL) devices have revolutionized glycemic control and the management of complex meals in patients with diabetes [15,16,17]. AHCL technology reduces blood glucose fluctuations and the risk of hypoglycemia by automatically modulating basal insulin levels and administering corrective boluses (every 60 min) when the glucose value detected by the CGM is predicted to exceed a predefined threshold [15,16]. However, the system is hybrid because the amount of ingested carbohydrates must be entered manually [15].

While there have been many experiments on pizza and glycemic control in T1D [17,18,19], there are no data on postprandial glucose levels in T1D children measured using CGM after eating rice cooked in different ways.

Since rice is always consumed after cooking, it is important to study the impact of this treatment on rice’s bioactive compounds. In our previous work, we investigated the impact of different cooking methods (boiling, microwave, pressure cooker, risotto) on Artemide black rice composition [2]. The boiling cooking method is the worst for preserving the healthy properties of this food, especially regarding vitamins and polyphenols. The cause is the elimination of the cooking water, in which some water-soluble polyphenols could be released, and thus lost. In this condition, a part of the soluble starch can also be removed, reducing the total carbohydrate content and increasing the fiber relative percentage.

Therefore, the primary aim of this study was to evaluate the chemical and nutritional composition of white rice before and after cooking (“risotto” method or boiling). The secondary aim was to study the glycemic impact of: (i) two different rice types (“Gigante Vercelli” white rice and Artemide black pigmented rice) and (ii) white rice cooked in two different ways (risotto and boiled) in T1D children and adolescents with or without CD using an AHCL system (Tandem Control-IQ^TM^).

## 2. Materials and Methods

### 2.1. Rice Samples

The “Artemide” black rice was supplied by the local company “Azienda Agricola Luigi e Carlo Guidobono Cavalchini, tenuta La Mondina”, located in Casalbeltrame, Novara (Italy), while the “Gigante Vercelli” white rice was supplied by the local farms “IDEAriso” soc.agr. srl, “Varalda” Fratelli Franco & Piergiorgio and “Tabacchi” di Massimo & Maurizio Tabacchi, located in Vercelli. Rice samples were provided in under vacuum packages kept at room temperature.

### 2.2. Chemical Characterization

All chemicals and reagents were of analytical grade and purchased from Merck KGaA (Darmstadt, Germany). Ultrapure water (18.2 MΩ cm at 25 °C) was produced using the ELGA PURELAB Ultra system (M-medical, Cornaredo, Milan, Italy).

Moisture, protein, and total dietary fiber were quantified following our previously published protocols on Artemide black rice [2]. Moisture content was determined using a thermo balance (MA30, Sartorius AG, Göttingen, Germany), while protein was quantified using the Kjeldahl method (conversion factor: 5.95) with the Kjeltec system I (Foss Tecator AB, Höganäs, Sweden). Total dietary fiber, defined as the complex mixture of nondigestible polymers (mainly polysaccharides), was determined following the Association of Official Agricultural Chemists (AOAC)-approved method 991-43 using the Megazyme Total Dietary Fiber Analysis kit (Megazyme, Wicklow, Ireland). Except for the moisture content, data are expressed as dry weight (d.w.).

The amounts of total, nonresistant (NR), and resistant (R) starch were determined using the respective assay procedures of Megazyme (K-DSTRS) kits based on the modified method of Englyst et al. [20] developed by McCleary [21]. Resistant starch (R) was defined as the proportion of starch indigestible in the upper gastrointestinal tract and is a component of dietary fiber.

Total polyphenols and antiradical activity were quantified using previously validated methods (Folin–Ciocalteu method and DPPH radicals, respectively) applying the procedures described in Colasanto et al. [2].

The rice cooking conditions are summarized in Table 1. Artemide black rice was cooked by boiling, while the Gigante Vercelli white rice was cooked both by boiling and using the risotto method.

For boiling, white or black rice and distilled water (in the ratio of 100 g rice/500 mL water) were placed in a steel pot and covered with a lid. The time required for proper cooking was calculated from the moment when the cooking water started boiling and was determined as 15 min for white rice and 35 min for pigmented rice. While cooking, the rice was occasionally mixed with a wooden spoon. At the end of cooking, the rice was drained and served to the enrolled subject.

For the risotto cooking method, white rice was placed in a cooking pan and toasted for 2 min, continuously mixing with a wooden spoon. Then, an aliquot of 50 mL of distilled water was added and cooking continued with occasional stirring. When the water was almost completely absorbed/evaporated, additional aliquots of 50 mL of water were added; the rice was left to cook for a final 15 min. The final ratio obtained was 100 g rice/350 mL water.

### 2.3. Subjects

The Ethical Committee of Novara approved the study (protocol number 108/2022), which conformed to the guidelines of the European Convention of Human Rights and Biomedicine for Research in Children. All patients were included in the study after they and their parents had provided written consent.

Subjects with T1D were enrolled at the Division of Pediatrics of Maggiore della Carità University Hospital in Novara, Italy in 2022. Two subjects also had CD diagnosed through routine screening approximately 1 year after the diagnosis of diabetes. Neither subject had symptoms of CD at diagnosis.

Children with diabetes were randomly recruited during routine clinic visits. The primary inclusion criteria were (i) therapy with AHCL (Tandem Control-IQ^TM^) for at least 3 months; (ii) aged between 10 and 16 years of age; and (iii) disease duration of at least 1 year.

Patients were excluded if they had an HbA1c > 8.5% (69 mmol/mol), food allergies, were treated with drugs that could affect immunity or glucose metabolism (corticosteroids, ciclosporin, tacrolimus), or had concurrent illness or psychiatric disease/eating disorders.

All subjects attended for 2 evenings to eat rice within a balanced meal established by a dietician (rice with zucchini, chicken, and fruit salad). During the 1st evening, children ate 80 g of white “Gigante Vercelli” rice (79.3 g CHO in 100 g rice). Four children ate rice cooked as “risotto”, and 4 subjects ate boiled rice. During the 2nd evening, all subjects received 80 g of boiled Artemide black rice (62 g CHO in 100 g rice) selected based on the results of the 1st evening.

Dinner was served at 7.30 p.m., prepared by the same cook, and was consumed after a standard insulin bolus delivered 10–20 min earlier according to glycemic values.

To standardize experimental conditions on each day, participants were instructed to do the following: on the morning before each study day, change the infusion set and sensor to ensure correct cannula positioning and optimal sensor accuracy; in the afternoon before dinner, not to play sports; before and after the meal, to correct any hypoglycemia or hyperglycemia according to the same standardized protocol.

All patients used the same short-acting insulin analog (insulin aspart, Novorapid, Novo Nordisk A/S, Bagsværd, Denmark).

Continuous glucose sensor data were collected from the Diasend platform for 12 h (7:00 p.m. to 7:00 a.m.) over the following day. Glucose values and times in target were evaluated every 5 min for 12 h after dinner. Total automatic basal insulin (U/kg) and total auto-bolus insulin (U/kg) infused during the entire observation period were evaluated.

### 2.4. Tandem Control-IQ^TM^ Technology

The Tandem Control-IQ technology is an advanced hybrid closed loop system approved in 2019 for T1D children older than 6 years that was released in Italy in 2020. The Tandem Control-IQ technology consists of the t:slim X2 insulin pump, which contains the Control-IQ algorithm (Tandem t:slim X2 with Control-IQ algorithm; San Diego, CA, USA) and works together with a continuous glucose monitor (CGM-Dexcom G6). The system automatically modulates basal insulin levels and performs corrective boluses (every 60 min) when the glucose values predicted by the CGM exceed a predefined threshold. Automatic corrections are allowed up to once per hour if necessary. However, the amount of ingested carbohydrates must be entered manually.

The Control-IQ technology allows the activation of functions such as Sleep and Exercise Activities that change the target range, allowing the algorithm to adapt to the different needs of the user. In sleep mode, the system does not deliver correction boluses and has a fixed target of 112.5–120 mg/dL.

There is also the possibility of using extended boluses with a maximum duration of 2 h to cover prolonged glycemic peaks after different types of meals. If the CGM connection has been lost for 20 min or longer, Control-IQ will stop automatically and the pump returns to the Personal Profile settings. Once the connection with CGM is restored, Control-IQ will resume automatically. 

### 2.5. Statistical Analysis

Statistical analyses were performed using SPSS v22.0 (SPSS Inc., Chicago, IL, USA) and R v4.2.2. Data were expressed as mean ± standard deviation (SD). Differences in rice composition were analyzed by analysis of variance (ANOVA) followed by Tukey’s honest significant difference test. Differences in continuous variables between groups were evaluated using the Mann–Whitney U test. The chi-squared test was used to compare nominal variables between groups. Trends in glycemic values were evaluated by multinomial regression analyses. A *p*-value less than 0.05 was considered significant.

## 3. Results

The gross composition of raw and cooked “Gigante Vercelli” white rice was determined, beside its antiradical activity, while the composition of Artemide black rice was derived from a previous work [2] and not reported in the present paper.

### 3.1. White Gigante Vercelli Rice: Chemical Characterization

The proximate composition, total phenolic content (TPC), and total antioxidant activity (TAA) of raw and cooked “Gigante Vercelli” white rice are shown in Table 2.

As expected, cooked rice had a significantly higher moisture content compared with raw samples, with no significant differences between risotto and boiled rice. There were no significant differences in protein content after cooking, but there was an increase in total dietary fiber after both cooking protocols (*p* < 0.05), probably due to the structural rearrangement of digestible starch into nondigestible/resistant starch, an important component of dietary fiber.

Boiling significantly reduced TPC and TAA (*p* < 0.01), while risotto cooking increased both TPC and TAA when compared with uncooked rice (*p* < 0.05). Thus, the risotto cooking not only preserved the antioxidant polyphenols with respect to boiling but also improved their content with respect to raw rice. These results can be explained by the production of Maillard compounds (presenting known antioxidant activity) during toasting (typical for risotto mode) as well as by the release of fiber-bound polyphenols in their free form.

Table 3 shows the starch composition of white rice. After boiling, there was a significant decrease in nonresistant (NR) starch and, consequently, total starch, while there were no significant differences in NR or total starch in risotto cooking compared with raw rice (*p* < 0.05). Both risotto cooking and boiling significantly increased resistant starch compared with raw samples (Table 3).

### 3.2. Postprandial Glucose Trends after Eating Risotto or Boiled Rice

We enrolled eight children with T1D, mean age 11 ± 1.4 years, disease duration 4.1 ± 3 years, and HbA1c 6.5 ± 0.4% (48 ± 0.4% mmol/mol). All children were of normal weight (BMI z-score −0.08 ± 0.6). Two subjects (25%) also had a diagnosis of CD. The children’s height, weight, and BMI were evaluated using Italian growth charts [22], and clinical data, HbA1c%, disease duration, and insulin requirements were collected.

There was a significant change in the glycemic trend (*p* < 0.0001) after consumption of risotto and boiled white rice (*p* = 0.02) over the following 12 h. Boiled black rice did not significantly alter the glycemic trend, with the mean glycemic peak never >180 mg/dL at each time point.

The mean glucose values at 2 (T2) and 4 (T4) hours were significantly lower after eating boiled white rice than after eating white risotto-cooked rice (*p* < 0.05). Moreover, subjects who ate boiled Artemide rice had significantly lower glucose levels at 2, 3, 4, and 5 h than those eating white risotto-cooked rice (*p* < 0.05) (Figure 1 and Appendix A). There were no differences in glycemic values between white and black boiled rice.

There were no significant differences in automatic basal or auto-bolus insulin deliveries by the AHCL according to different types of rice or cooking method. There were also no differences between subjects with or without CD.

## 4. Discussion

Rice is widely consumed around the world, and its intake in diabetes is not easy to manage as it depends on several factors such as glycemic index and cooking method. Moreover, rice is a principal gluten-free grain for people with CD.

Brown and white rice have different GIs and, consequently, a different impact on glucose levels in people with diabetes [1]. Different cooking techniques have been shown to influence the composition, digestibility, and GI of rice [1,23]. For example, raw or partially cooked starch can be considered a low-GI food. The ingestion of cooked white rice cooled to 4 °C for 24 h and then reheated produced a lower glycemic response than the ingestion of freshly cooked white rice. Therefore, rice preparation has a significant impact on glycemic variability [24], as observed for other foods. Starch structure, gelling, or retrogradation status (as well as the ratio between amylose and amylopectin) produce different digestive profiles, thus, modifying the GI. More particularly, gelatinization and retrogradation effects on starch can modulate the effective digestion, according to their thermal treatments and cooking mode. Retrograded rice has higher resistant starch levels when compared with nonretrograded rice. The presence of residual crystallites may reduce the physical accessibility of the digestion enzymes; thus, reducing the digestibility of starch has an impact on GI. Moreover, partially gelatinized starches exhibit more resistance to enzymatic digestion than do the retrograded starches [25].

We found no significant differences in the protein content of white rice after cooking, but we did detect an interesting increase in total dietary fiber. TPC and TAA also both increased after cooking using the risotto method and decreased after boiling.

We previously reported the composition of Artemide rice [2]. Artemide black rice similarly showed no significant differences in protein content after both types of cooking (average 10.4% vs. 10.5% for raw rice), while total dietary fiber significantly decreased compared with raw rice (7.9 vs. 10.8%), probably because cooking led to the partial degradation and release of some fiber components.

The polyphenol content and antiradical activity of Artemide black rice were better preserved after risotto cooking (21% of the total polyphenol content), while boiling preserved only 10%, making the risotto method the healthiest approach for cooking Artemide black rice [2]. However, in practice, Artemide black rice is very difficult to cook risotto style due to its typical brown rice-related characteristics.

The results were similar for white “Gigante Vercelli” rice, although the polyphenol concentration was much lower than in Artemide rice (51.8 mg/g dry weight raw rice for Artemide vs. 0.205 mg/g dry weight raw rice for “Gigante Vercelli”). Nevertheless, white rice is easy to prepare risotto style and is very palatable.

AHCL represents a significant advance in diabetes management [15,26] for postprandial glycemic control and the management of complex meals. We observed better postprandial glycemic control after eating boiled white rice than risotto-cooked rice and an important change in glycemic trend after eating white rice, especially when cooked as risotto. These results suggest that the type of cooking has a significant impact on the GI and glycemia. We measured these differences through AHCL sensor values, which suggested that some soluble starch is lost in the cooking water by boiling. Risotto cooking allows all the cooking water to be absorbed by the rice to produce the higher glycemic values found in the risotto group.

Eating boiled Artemide black rice resulted in better glycemic control at each time point than eating white rice, with mean values always in a target range under 180 mg/dL. This is probably because Artemide black rice, being unmilled, has a higher fiber content and, consequently, lower starch concentration than “Gigante Vercelli” white rice.

The main limitation of this study was the small number of children enrolled. However, this remains an important pilot study that provides new data on the chemical and nutritional characteristics of meal-based rice and how the type of rice and cooking impact glucose levels. We were unable to cook Artemide risotto style for comparison because it is too difficult to prepare. Finally, we were unable to evaluate glycemic differences between diabetics with and without CD due to the sample size.

## 5. Conclusions

Risotto is an important dish in Italian cuisine, safe for people with CD, and can be more palatable than boiled rice. However, risotto-cooked white rice is the most difficult to manage in terms of postprandial blood glucose control in patients with diabetes, even though this cooking mode has resulted in a slight increase in total polyphenols and antioxidant activity. Nevertheless, the improved knowledge of how different types of rice and cooking methods influence glycemia could be exploited to clarify or understand metabolic outcomes. The phenomenon of gelatinization of starch is particularly correlated with the digestibility of rice and strictly depends on the cooking mode preparation, as previously reported in the scientific literature. Black Artemide rice provides the best postprandial glycemic values, while white Gigante Vercelli rice produces a lower glycemic peak if cooked by boiling than risotto-style. This pilot study provides the rationale for further clinical studies to evaluate the influence of the advanced hybrid closed-loop functions on glycemia after eating difficult-to-manage dishes such as white risotto rice. Extended insulin boluses to cover glycemic peaks administered by advanced technology might help people with diabetes obtain better postprandial glycemic control when eating meals with high glycemic indices or cooked in styles that promote glycemic peaks.

## Figures and Tables

**Figure 1 nutrients-15-01654-f001:**
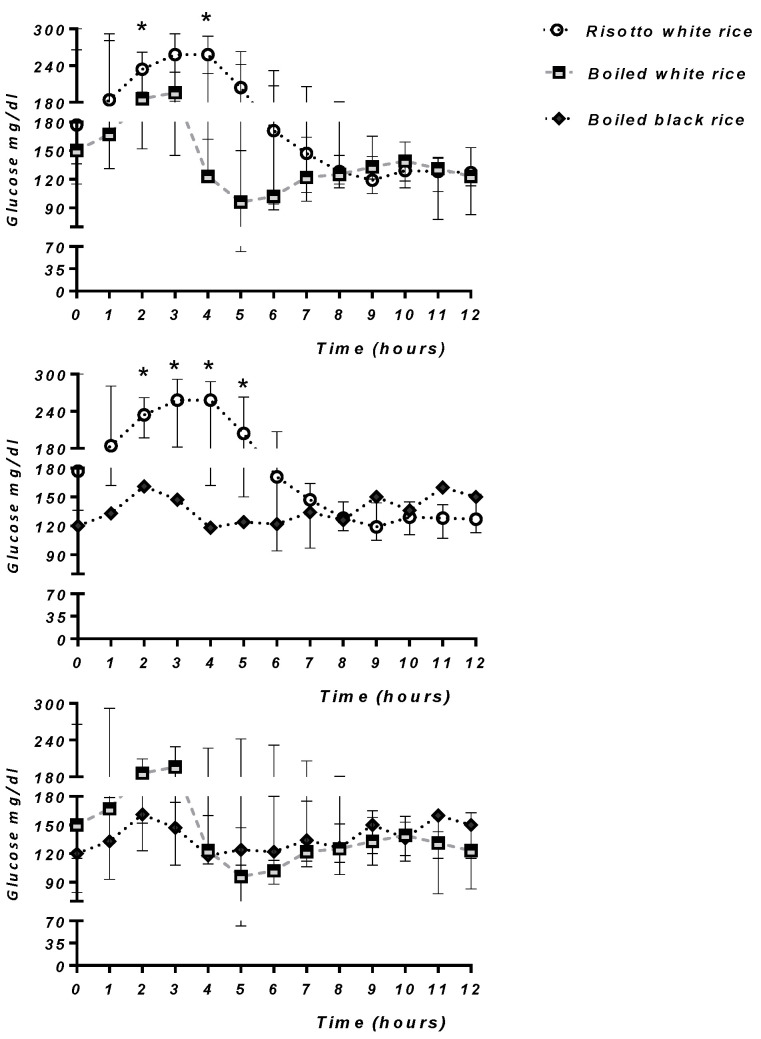
Glycemic trends after consumption of risotto white rice, boiled white rice, and boiled black rice over the following 12 h. * *p* < 0.05.

**Table 1 nutrients-15-01654-t001:** The rice cooking conditions used in the study.

	Cooking Method	Rice: Water Ratio (g/mL)	Cooking Time (min)	Notes
Gigante Vercelli	Boiling (BOI)	100/500	15	Rice added after the water had started boiling
	Risotto (RIS)	100/350	15	Time including 2 min of toasting
Artemide	Boiling (BOI)	100/500	35	Rice added after the water had started boiling

**Table 2 nutrients-15-01654-t002:** Proximate composition, total phenolic content (mg CE/g d.w.), and total antioxidant activity (mg TE/g d.w.) determined in raw and cooked “Gigante Vercelli” white rice.

Sample	Moisture (%)	Proteins (% d.w.)	Total Dietary Fiber (% d.w.)	Total Polyphenols (mg CE/g d.w.)	Antiradical Activity (mg TE/g d.w.)
Raw	10.8 ± 0.4 ^b^	7.02 ± 0.3 ^a^	1.14 ± 0.1 ^b^	0.21 ± 0.02 ^b^	0.20 ± 0.01 ^b^
Risotto	65.5 ± 0.2 ^a^	7.49 ± 0.3 ^a^	2.01 ± 0.2 ^a^	0.24 ± 0.01 ^a^	0.25 ± 0.01 ^a^
Boiled	65.4 ± 0.7 ^a^	7.39 ± 0.1 ^a^	1.79 ± 0.04 ^a^	0.09 ± 0.01 ^c^	0.10 ± 0.01 ^c^

Results are expressed as mean ± SD. Values with different letters in the same column are significantly different (a: not significant; b: *p* < 0.05; c: *p* < 0.01).

**Table 3 nutrients-15-01654-t003:** Nonresistant (NR), resistant (R), and total starch in raw and cooked “Gigante Vercelli” white rice.

Sample	NR Starch	R Starch	Total Starch
**Raw**	84.6 ± 1.2 ^a^	0.22 ± 0.1 ^b^	84.8 ± 1.3 ^a^
**Risotto**	84.8 ± 1.1 ^a^	0.80 ± 0.1 ^a^	85.6 ± 1.2 ^a^
**Boiled**	79.1 ± 2.3 ^b^	0.77 ± 0.03 ^a^	79.9 ± 2.3 ^b^

Results are expressed as mean ± standard deviation. Values with different letters in the same column are significantly different (*p* < 0.05).

## Data Availability

Not applicable.

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
