# Peer review of "The Impact of Different Types of Rice and Cooking on Postprandial Glycemic Trends in Children with Type 1 Diabetes with or without Celiac Disease"

_nutrients, 2023, doi:10.3390/nu15071654_

Round 1

Reviewer 1 Report

This study compares nutritional composition of one type of white rice cooked in two different ways (boiled and risotto) and one type of brown rice boiled. Next to this description of composition, a clinical study has been performed in 8 Type 1 diabetes patients, of whom 2 have concomittant coeliac disease. 

Before the paper being ready for publications, several point need clarification. Please explain what you aim to express in line 46-47, about the relation between glutenfree diet and glycemic control. Also in the introduction, explain in more detail how AHCL systems function. The aim of the study makes a difference between type1 diabetes children with and without celiac disease, but in the results nor in the conclusions is this difference discussed. What is the relevance? It seems very unlikely to make meaningful comparisons between 6 T1D children without, and 2 with celiac disease. As for the method section, pleas explain how fibers have been defined and how has the content been analysed? What was the difference between fiber and resistant starch? For how long were the patients with celiac disease on a glutenfree diet (this is important in order to have an idea whether the small bowel mucosa will have healed). How was the diagnosis suspected (asymptomatic diagnosed following screening? otherwise?). Did diagnosis of CD follow diagnosis of T1D or vice versa?

Subject: please do not mention results in the method section, but instead specify both inclusion and exclusion criteria (now exclusion criteria are mentioned in line 106-111, but no clear inclusio criteria are set. Description of the final study cohort is part of the result section.

How was the protocol conceived? The authors mention that Artemide rice was only presented in boiled version because of the results of the first evening where half of the children did receive boiled and half risotto white rice. Had this been foreseen in the protocol? Or was Artemide only presented in boiled version because of the fact that it is difficult to prepare as risotto? line 93: please detail what kind of bolus was given before the meal (probably "insulin", if so: mention). Line 106-111 are better placed directly following inclusion criteria in section 2.2. 

Some of the results are intriguing and not further discussed. How do you explain a rise in fiber content (expressed per dry weight) cfr line 138? It is not clear how line 188-191 related to the results. Please restructure the discussion: what are the results, how do these relate to what is known, to other findings in the litterature? What are the strengths and limitations of this study? What are the main conclusions? suggestions for further research.

Please limit the conclusions to what can be based on these results. Line 256-258 refers to an evaluation of the AHCL system, which was not the goal of this study, neither the experimental design (as all children had such a system, there is no comparison to other systems of glycemic control).

Finally, please ensure the paper is corrected by a native speaker. some examples: Line 90: ...only in the boiled version (and not: in the only boiled version). Line 94: to standardise (instead of "uniform"). line 117: Mann-Whitney U test

Author Response

March 16, 2023
To whom it may concern:
Re. The impact of different types of rice and cooking on postprandial glycemic trends in children with type 1 diabetes with or without celiac disease
Many thanks to the editor and reviewers for considering this manuscript and for the very valuable feedback. We have now addressed these points in full, and point-by-point responses are appended below. Major changes are highlighted in red in the revised version of the text.
Furthermore, the entire document has been thoroughly edited by a native English speaker.
We hope that you agree that the manuscript is now significantly improved, and we hope that it is now suitable for publication in Nutrients.
Yours sincerely

Point-by-point responses

This study compares nutritional composition of one type of white rice cooked in two different ways (boiled and risotto) and one type of brown rice boiled. Next to this description of composition, a clinical study has been performed in 8 Type 1 diabetes patients, of whom 2 have concomittant coeliac disease. 

Before the paper being ready for publications, several point need clarification.

Please explain what you aim to express in line 46-47, about the relation between glutenfree diet and glycemic control.

Thanks for your comment. While some studies have shown that a GFD can have a positive impact on growth, data on the impact of a GFD on glycemic control, HbA1c, and hypoglycemic episodes in subjects with T1D and CD are conflicting. A low glycemic index diet can certainly be useful for reducing glucose excursions, but most GFD foods have a high glycemic index that can rapidly raise blood glucose levels and risk major glycemic variability. We have now explained this in the Introduction, lines 48-53.  

Also in the introduction, explain in more detail how AHCL systems function.

We have now added a brief description on how AHCL systems function in the Introduction, lines 61-65. Moreover, we added a description of the Tandem Control system to the Methods (lines 112-114).

The aim of the study makes a difference between type1 diabetes children with and without celiac disease, but in the results nor in the conclusions is this difference discussed. What is the relevance? It seems very unlikely to make meaningful comparisons between 6 T1D children without, and 2 with celiac disease.

Thanks, and you are of course correct. Our primary aim was to study the chemical composition of white rice according to the method of cooking (risotto vs boiling), and the secondary aim was to evaluate the glycemic impact of the two different types of rice and cooking method in T1D children and adolescents with or without CD using an AHCL system (Tandem Control-IQ). We did not aim to compare children with and without CD, as this evaluation was not possible due to the small number of subjects. This was a pilot study to pave the way for a larger clinical study capable of evaluating the influence of the AHCL systems on difficult-to-manage foods such as rice.  

As for the method section, pleas explain how fibers have been defined and how has the content been analysed? What was the difference between fiber and resistant starch?

We have now added the methods for fiber quantification and the definition of dietary fiber (lines 82-85). The resistant starch is a component of dietary fiber not hydrolyzed by human amylases.

For how long were the patients with celiac disease on a glutenfree diet (this is important in order to have an idea whether the small bowel mucosa will have healed). How was the diagnosis suspected (asymptomatic diagnosed following screening? otherwise?). Did diagnosis of CD follow diagnosis of T1D or vice versa?

We have added this information to the Methods: “Two subjects also had a diagnosis of CD diagnosed through routine screening approximately one year after the diagnosis of diabetes. Neither subject had symptoms of CD at diagnosis”, lines 106-108.

Subject: please do not mention results in the method section, but instead specify both inclusion and exclusion criteria (now exclusion criteria are mentioned in line 106-111, but no clear inclusion criteria are set). Description of the final study cohort is part of the result section.

We have now better defined the inclusion and exclusion criteria in the Methods (lines 109-118), and we have focused the description of the final study cohort in the results, as suggested.

How was the protocol conceived? The authors mention that Artemide rice was only presented in boiled version because of the results of the first evening where half of the children did receive boiled and half risotto white rice. Had this been foreseen in the protocol? Or was Artemide only presented in boiled version because of the fact that it is difficult to prepare as risotto?

Your latter assumption is correct. We wanted to study the impact of different types of rice and cooking on glycemic values, but Artemide rice cannot be prepared as risotto without pre-boiling. We have clarified this (lines 252-253).

line 93: please detail what kind of bolus was given before the meal (probably "insulin", if so: mention).

Thanks, it was an insulin bolus.

Line 106-111 are better placed directly following inclusion criteria in section 2.2. 

Thanks, we have moved this.

Some of the results are intriguing and not further discussed. How do you explain a rise in fiber content (expressed per dry weight) cfr line 138? It is not clear how line 188-191 related to the results. Please restructure the discussion: what are the results, how do these relate to what is known, to other findings in the litterature? What are the strengths and limitations of this study? What are the main conclusions? suggestions for further research.

We have now focused our discussion in the context of our results and the literature. In particular, the increase in fiber content could be explained by the structural rearrangement of digestible starch during processing/cooking into resistant/non-digestible starch, such that this component is then included as dietary fiber. We have clarified this on lines 161-163.

We have also updated the limitations and the conclusions. Furthermore, the thorough editing has improved the narrative throughout.

Please limit the conclusions to what can be based on these results. Line 256-258 refers to an evaluation of the AHCL system, which was not the goal of this study, neither the experimental design (as all children had such a system, there is no comparison to other systems of glycemic control).

We have removed lines 256-258, as suggested, and better structured the conclusion.

Finally, please ensure the paper is corrected by a native speaker. some examples: Line 90: ...only in the boiled version (and not: in the only boiled version). Line 94: to standardise (instead of "uniform"). line 117: Mann-Whitney U test

The paper has now been thoroughly edited by a native English speaker.

Reviewer 2 Report

The authors of this manuscript describe an interesting study done to characterize rice and study the postprandial glycemic response. Although interesting work, the submitted manuscript seems to be missing several key aspects that must be improved before suitability for publication could be determined.

Abstract: Revise the first sentence (line 11). Rephrase for clarity. For example, something like Rice is widely consumed and popular for individuals with celiac disease, however it can be problematic for individuals with diabetes as it presents a high glycaemic index.

Introduction:

Lines 51-52 are misleading. It sounds like the authors are suggesting that there are no studies that have looked at the effects of cooking or processing on rice and postprandial glucose levels. There appears to be many. Perhaps the authors are trying to identify a specific question that is still not clear amongst the past studies that have been done?  

Lines 57-69 have a separate paragraph to discuss AHCL. Perhaps, the authors need to include a bit more information about this. For example, defining AHCL and why it is of great help.

Materials and Methods

The authors should consider including a section at the beginning of the Materials and Methods section to identify the major reagents/materials they purchased and the suppliers

Revise “Chemical characterizations: techniques” as it is repetitive to have both characterizations and techniques. Also, characterization does not need to be plural here.

Although lines 68-76 reference other papers for methods, it would be good to very briefly mention a few key details about each chosen method. For example, for protein content, clarifying what method was used. The rice cooking method should be further clarified. I assume this was with a pot and not a rice cooker? Note, the notes for the Gigante Vercelli RIS are not clear (when was the rice added?).

The authors should include their protocol approval (e.g., institutional review board, ethics, etc.).

The authors should clarify the criteria for the participants/subjects to be able to participate in the study and how they were recruited. It’s also not clear why 8 subjects were used (e.g., power analysis or some other reason?).

Typo line 85.

Line 87 mentions four “children” but in other places the participants are described as subjects. A better description of the subjects should be included.

The materials and methods section needs a bit of work to improve the clarity and to ensure that it is clear for the reader.

Two types of rice are described in the materials and methods section, but there only appears to be one type (Gigante Vercelli) shown in the tables (2 and 3).

 There are a number of measurements that are briefly described in the materials and methods section regarding the human study portion, however, only Figure 1 shows data related to the human studies. Perhaps the authors need to include additional data in a table or figure.

Author Response

March 16, 2023

To whom it may concern:

Re. The impact of different types of rice and cooking on postprandial glycemic trends in children with type 1 diabetes with or without celiac disease

Many thanks to the editor and reviewers for considering this manuscript and for the very valuable feedback. We have now addressed these points in full, and point-by-point responses are appended below. Major changes are highlighted in red in the revised version of the text.

Furthermore, the entire document has been thoroughly edited by a native English speaker.

We hope that you agree that the manuscript is now significantly improved, and we hope that it is now suitable for publication in Nutrients.

Yours sincerely

Point-by-point responses

The authors of this manuscript describe an interesting study done to characterize rice and study the postprandial glycemic response. Although interesting work, the submitted manuscript seems to be missing several key aspects that must be improved before suitability for publication could be determined.

Abstract: Revise the first sentence (line 11). Rephrase for clarity. For example, something like Rice is widely consumed and popular for individuals with celiac disease, however it can be problematic for individuals with diabetes as it presents a high glycemic index.

Thank you, we have modified the abstract.

Introduction:

Lines 51-52 are misleading. It sounds like the authors are suggesting that there are no studies that have looked at the effects of cooking or processing on rice and postprandial glucose levels. There appears to be many. Perhaps the authors are trying to identify a specific question that is still not clear amongst the past studies that have been done?  

We agree that there are some studies on rice and glycemic values in diabetes, but no study has evaluated postprandial glucose values with continuous glucose monitoring after eating rice cooked in different ways and fed to T1D children. We now clarify this in the revised manuscript (lines 65-67).  

Lines 57-69 have a separate paragraph to discuss AHCL. Perhaps, the authors need to include a bit more information about this. For example, defining AHCL and why it is of great help.

We have now added a brief description on how AHCL systems function to the Introduction, lines 61-65. Moreover, we added a description of the Tandem Control system to the Methods (lines 112-114).

Materials and Methods

The authors should consider including a section at the beginning of the Materials and Methods section to identify the major reagents/materials they purchased and the suppliers

We have added the sentence: “All chemicals and reagents were of analytical grade and purchased from Merck KGaA (Darmstadt, Germany)” on lines 76-78.

Revise “Chemical characterizations: techniques” as it is repetitive to have both characterizations and techniques. Also, characterization does not need to be plural here.

Thanks. We have modified this to “Chemical characterization”, as suggested.

Although lines 68-76 reference other papers for methods, it would be good to very briefly mention a few key details about each chosen method. For example, for protein content, clarifying what method was used. The rice cooking method should be further clarified. I assume this was with a pot and not a rice cooker? Note, the notes for the Gigante Vercelli RIS are not clear (when was the rice added?).

We have now added brief descriptions of the analytical procedures and clarified the cooking methods used (lines 80-98).

The authors should include their protocol approval (e.g., institutional review board, ethics, etc.).

Thanks. The study was approved by the Ethical Committee of Novara (protocol number 108/2022). We have added this to the text (lines 101-104).

The authors should clarify the criteria for the participants/subjects to be able to participate in the study and how they were recruited. It’s also not clear why 8 subjects were used (e.g., power analysis or some other reason?). Typo line 85. 

We have now clarified the inclusion criteria in the revised Methods (lines 109-114), and we have focused on the description of the final study cohort in the results. This was a small, pilot study as a prequel to a larger definitive study, hence the small number of patients. We have clarified this.   

Line 87 mentions four “children” but in other places, the participants are described as subjects. A better description of the subjects should be included. 

Thanks, we have fully described the inclusion and exclusion criteria and the description of the subjects to the Methods (lines 109-114) and Results (lines 185-189). All subjects in our study were children aged between 10 and 16 years with type 1 diabetes.

The materials and methods section needs a bit of work to improve the clarity and to ensure that it is clear for the reader.

Thanks, we have revised the Methods for clarity.

Two types of rice are described in the materials and methods section, but there only appears to be one type (Gigante Vercelli) shown in the tables (2 and 3). 

Thank you. Table 2 only presents results for white rice because the ones related to the black rice variety are not significantly different to those reported in our previous study [Colasanto et. al 2021]. The results on starch content (Table 3) refer only to the white variety because we wanted to compare differences in starch composition after different cooking methods, which was only possible for the white variety because black rice cannot be cooked in the risotto style.

There are a number of measurements that are briefly described in the materials and methods section regarding the human study portion, however, only Figure 1 shows data related to the human studies. Perhaps the authors need to include additional data in a table or figure.

Thanks. Glucose values were evaluated every 5 minutes for 12 hours after dinner through continuous glucose monitoring. We have now added a table reporting the glucose values (mean ± SD) to the Supplementary. In particular, glucose values were evaluated as the average values of the 12 measurements recorded each hour.

Round 2

Reviewer 1 Report

The authors have well responded to the comments on the first version. The paper is more clearly structured. No major comments. The introduction has been expanded substantially, with not all information on T1D and CD being important for this study, so one can wonder whether this section should be shortened and more focused on what is important for this study (by limiting the information on T1D and CD).

Author Response

Many thanks to the editor and reviewers for your positive comments.

We have revised the manuscript, and point-by-point responses are appended below. Our changes are highlighted in red in the revised version of the text.

We hope that it is now suitable for publication in Nutrients.

Yours sincerely

The authors have well responded to the comments on the first version. The paper is more clearly structured. No major comments.

Thank you for your positive comment and for the constructive suggestions provided.

The introduction has been expanded substantially, with not all information on T1D and CD being important for this study, so one can wonder whether this section should be shortened and more focused on what is important for this study (by limiting the information on T1D and CD).

Thanks for this observation. We have expanded the introduction and added information on diabetes and celiac disease because other reviewers asked for it. However, we have shorted this point as suggested.

Reviewer 2 Report

The authors have made significant improvements to their manuscript and have improved clarity with the additions.

Lines 365-367, do the authors mean that the risotto rice method better preserves polyphenol and antioxidant activity compared to boiled rice? I assume so, but a minor adjustment in the sentence could clarify this.

Lines 367-368 - I think I understand what the authors mean, but the sentence should be revised as it is not currently complete.

An explanation was added (lines 269-271), but lines 272-273 could benefit from an explanation of the results.

Author Response

March 24, 2023

Many thanks to the editor and reviewers for your positive comments.

We have revised the manuscript, and point-by-point responses are appended below. Our changes are highlighted in red in the revised version of the text.

We hope that it is now suitable for publication in Nutrients.

Yours sincerely

Rev 2

The authors have made significant improvements to their manuscript and have improved clarity with the additions.

Thank you for your positive comment and for the constructive suggestions provided.

Lines 365-367, do the authors mean that the risotto rice method better preserves polyphenol and antioxidant activity compared to boiled rice? I assume so, but a minor adjustment in the sentence could clarify this.

Thanks for your observation. The risotto rice method is the most difficult to manage in terms of good glycemic control in our patients, but it better preserves polyphenol and antioxidant activity vs boiled rice. We have modified the sentences as suggested 

Lines 367-368 - I think I understand what the authors mean, but the sentence should be revised as it is not currently complete.

Thank you for your comment. We have modified the conclusion and revised our sentence.

An explanation was added (lines 269-271), but lines 272-273 could benefit from an explanation of the results.

Thank you for your comment. We have better explained our result, as suggested.
